# dsRNA-Seq: Identification of Viral Infection by Purifying and Sequencing dsRNA

**DOI:** 10.3390/v11100943

**Published:** 2019-10-14

**Authors:** Carolyn J. Decker, Halley R. Steiner, Laura L. Hoon-Hanks, James H. Morrison, Kelsey C. Haist, Alex C. Stabell, Eric M. Poeschla, Thomas E. Morrison, Mark D. Stenglein, Sara L. Sawyer, Roy Parker

**Affiliations:** 1Department of Biochemistry, University of Colorado, Boulder, CO 80303, USA; carolyn.decker@colorado.edu (C.J.D.); halley.steiner@colorado.edu (H.R.S.); 2Howard Hughes Medical Institute, University of Colorado, Boulder, CO 80303, USA; 3Department of Microbiology, Immunology, and Pathology, Colorado State University, Fort Collins, CO 80523, USA; laura.hoon-hanks@colostate.edu (L.L.H.-H.); mark.stenglein@colostate.edu (M.D.S.); 4Division of Infectious Diseases, University of Colorado School of Medicine, Aurora, CO 80045, USA; james.morrison@cuanschutz.edu (J.H.M.); eric.poeschla@cuanschutz.edu (E.M.P.); 5Department of Immunology and Microbiology, University of Colorado School of Medicine, Aurora, CO 80045, USA; kelsey.haist@cuanschutz.edu (K.C.H.); thomas.morrison@cuanschutz.edu (T.E.M.); 6Department of Molecular, Cellular and Developmental Biology, University of Colorado, Boulder, CO 80302, USA; alexstabell@gmail.com (A.C.S.); ssawyer@colorado.edu (S.L.S.)

**Keywords:** dsRNA, double-stranded RNA, emerging disease, emerging viruses, RNA virus, RNA-Seq

## Abstract

RNA viruses are a major source of emerging and re-emerging infectious diseases around the world. We developed a method to identify RNA viruses that is based on the fact that RNA viruses produce double-stranded RNA (dsRNA) while replicating. Purifying and sequencing dsRNA from the total RNA isolated from infected tissue allowed us to recover dsRNA virus sequences and replicated sequences from single-stranded RNA (ssRNA) viruses. We refer to this approach as dsRNA-Seq. By assembling dsRNA sequences into contigs we identified full length or partial RNA viral genomes of varying genome types infecting mammalian culture samples, identified a known viral disease agent in laboratory infected mice, and successfully detected naturally occurring RNA viral infections in reptiles. Here, we show that dsRNA-Seq is a preferable method for identifying viruses in organisms that don’t have sequenced genomes and/or commercially available rRNA depletion reagents. In addition, a significant advantage of this method is the ability to identify replicated viral sequences of ssRNA viruses, which is useful for distinguishing infectious viral agents from potential noninfectious viral particles or contaminants.

## 1. Introduction

RNA viruses have an enormous impact on human health and constitute a major source of emerging or re-emerging infectious diseases [1], such as those caused by MERS, Ebola, West Nile, Zika, and chikungunya viruses [2,3]. RNA viruses also pose a threat to animal and plant health, where they can cause major loss to crop and animal production and impact biodiversity [4,5,6]. In order to curb emerging infectious agents, it is useful to have robust methods to identify new viruses. Additionally, the threats of synthetic or otherwise bioengineered viruses present serious challenges to modern approaches to viral identification, since such viruses need not share any similarity to previously known viral agents [7]. To reduce morbidity and mortality, robust methods to identify new viruses are needed.

High-throughput sequencing of total RNA isolated from infected individuals is a powerful approach to identifying and determining complete genomes of RNA viruses that are potentially responsible for diseases without a known causative agent. In addition, this approach allows for the detection of variants of known viruses or synthetic viral agents, which may not be recognized by PCR or serological-based techniques. One limitation is that viral sequences are present at very low levels relative to host sequences in clinical samples, which limits the sensitivity of viral detection and the ability to reconstruct viral genomes [8,9,10,11].

Several current approaches enrich for viral sequences in order to improve the sensitivity of detection. Because the vast majority of host RNA is ribosomal RNA, host rRNA depletion is often used [9,12]. Another approach is to enrich for viral particles from infected samples [13], the feasibility of which is dependent on the sample type and whether sufficient viral particles can be obtained. Positive selection strategies have also been developed, wherein viral sequences are “captured” by hybridization to virus-specific probe sets based on known viruses [14,15,16]. However, these strategies may be biased against detecting novel viruses, depending on how closely they are related to known viruses.

We have developed an alternative method to enrich for viral RNA sequences that incorporates both negative selection to remove host RNA and positive selection for dsRNA viruses and dsRNA replication intermediates of ssRNA viruses. This method is not based on known viral sequences. We reasoned that all RNA viruses produce dsRNA while replicating, therefore by purifying dsRNA from the total RNA isolated from infected cells or tissues we can enrich for viral RNA sequences. In the case of single-stranded RNA viruses, this approach would also allow us to distinguish nonreplicating viral particles from replicating infectious agents. High throughput sequencing of dsRNA has been used previously to identify RNA viruses, primarily in plants [17,18,19,20], but also in fungi [21], microbial communities [22,23], and a deep-sea tube worm [24]. Here, we adapted this approach to the identification of animal viruses. We developed a stringent method to purify dsRNA. We first remove the majority of host RNA by treating with a single-strand specific RNase, and then isolate the dsRNA by immunoprecipitation with a sequence-independent anti-dsRNA antibody [25,26]. We then sequence and de novo assemble the resulting dsRNA to aid in viral discovery. We refer to this approach as dsRNA-Seq and have used it successfully to identify a variety of RNA viruses in infected animal tissues.

## 2. Materials and Methods

### 2.1. dsRNA Purification

Total RNA from Vero cells (CCL-81, ATCC, Mannassas, VA, USA) infected with dengue virus type 2 (New Guinea C), influenza A (H3N2 Udorn), or mock-infected was extracted using Trizol (Thermo-Scientific, Waltham, MA, USA) [27]. The total RNA extracted from whole quad muscle collected five days post-infection from C57BL/6J mice that were mock-infected or infected with Ross River virus (T48), as described in [28], was provided by Kelsey Haist and Thomas Morrison (University of Colorado, Anschutz Medical Campus). All mouse studies were performed at the University of Colorado Anschutz Medical Campus (Animal Welfare Assurance #A 3269-01) using protocols approved by the University of Colorado Institutional Animal Care and Use Committee and in accordance with the recommendations in the Guide for the Care and Use of Laboratory Animals of the National Institutes of Health. Total RNA extracted from green tree python lung and pooled lung/esophagus, rough-scaled python lung, mule deer brain and lymph node, boa constrictor kidney, and veiled chameleon pooled lung/trachea/oral mucosa and 2 samples of pooled lung/liver/kidney as described in [29], was provided by Laura Hoon-Hanks and Mark Stenglein (Colorado State University). All of the reptile and deer samples were collected postmortem from client-owned animals for diagnostic assessment. dsRNA was purified from 100 µg total RNA isolated from Vero cells, 5 µg total RNA from mouse skeletal muscle samples, and 10 µg total RNA from reptilian and mule deer samples. Total RNA, at final concentration of 0.2 µg/µL, was incubated with 1 unit RNase 1 (Ambion, Waltham, MA, USA) (10 units RNase 1 for reptilian and mule deer samples) and 0.2 units Turbo DNase 1 (Ambion, Waltham, MA, USA) per µg total RNA in 1× Turbo DNase 1 buffer (which contains 75 mM monovalent salt) and 125 mM NaCl (final monovalent salt concentration 0.2 M) at 37 °C for 30 min. Reactions were then diluted with buffer pre-chilled on ice to a final concentration of 20 mM TrisCl pH 7.5, 0.15 M NaCl, 0.2 mM EDTA, 0.2% Tween20 to a final volume of 500 µL for mouse, reptile and deer samples or 1 mL for Vero cell samples, and then incubated with 5 µg of J2 anti-dsRNA antibody (Scicons, Szirák, Hungary) [25] pre-bound to 0.75 µg of Protein A Dynabeads (Invitrogen, Waltham, MA, USA) with end to end rotation at 4 °C for 2 h. Beads were recovered using a magnet and washed three times with 1× IP buffer (20 mM TrisCl pH 7.5, 0.15 M NaCl, 0.1 mM EDTA, 0.1% Tween20). The dsRNA was recovered from the beads by adding 150 µL 1× IP buffer and 450 µL of Trizol LS (Ambion, Waltham, MA, USA) and then following the manufacturer’s protocol for isolating RNA. The resulting aqueous phase was mixed with equal volume 70% ethanol, applied to RNA Clean-Up and Concentration Micro-Elute columns (Norgen Biotek, Thorold, ON, Canada) following manufacturer’s protocol and the dsRNA eluted in 15 µL of nuclease-free water.

### 2.2. RNA Library Construction and Sequencing

For Vero cell culture samples, 9 µL dsRNA in water was denatured at 95 °C for 2 min and then cooled on ice. RNA libraries were then prepared using ScriptSeq v2 Stranded Kit (Epicentre, Madison, WI, USA) following the manufacturer’s protocol, except fragmentation was done at 85 °C for 8 min. Each sample was indexed with a 6 bp unique barcode, libraries were pooled and sequenced on MiSeq (75 base PE reads, Illumina, San Diego, CA, USA). For mouse, reptilian, and mule deer tissue samples, 11 µL dsRNA diluted to final 10 mM TrisCl pH 8.0, 0.1% Tween 20 in 12 µL was denatured at 95 °C for 2 min, then cooled on ice. RNA libraries were then prepared using Ovation SoLo RNA-Seq System (NuGen, Redwood City, CA, USA), starting at Step C and stopping after Step L (Library Amplification I Purification) in the manufacturer’s protocol; therefore, the dsRNA libraries did not undergo the ribosomal sequence depletion steps in the protocol. RNA libraries were also prepared from 2 ng of total RNA from the mouse samples using the Ovation SoLo RNA-Seq System (NuGen, Redwood City, CA, USA) following the manufacturer’s protocol, starting at Step C, including the ribosomal sequence depletion steps, and stopping after Step P (Library Amplification II purification). Each sample was indexed with an 8 bp unique barcode, libraries were pooled and sequenced on NextSeq (75 base PE reads, Illumina, San Diego, CA, USA) using SoLo Custom R1 primer and standard Illumina R2 primer.

### 2.3. Analysis of dsRNA-Seq Sequences from Vero Cell Samples

Illumina adaptors were trimmed in paired end (PE) mode, reads trimmed when the average quality score in four base windows fell below 20, and then reads smaller than 50 nt were discarded using Trimmomatic 0.32 [30]. Reads were assembled into contigs 500 nt or longer using Trinity 2.0.6 [31] and the longest isoform of related contigs was selected using the Trinity helper script get_longest_isoform_seq_per_trinity_gene.pl. Contigs were mapped to the *Chlorocebus sabaeus* genome (GCF_000409795.2), dengue virus type 2 (NC_001474.2), and influenza A virus (NC_007366.1) genomes using the bwa-mem algorithm of bwa 0.7.15 [32]. To look for similarities between the contigs and known sequences the contigs were used as queries using the BLASTn command of BLAST 2.7.1 [33] against the NCBI nt database. Contigs with similarity to phiX174 were removed. Bowtie 2.2.9 [34] was used to map the dsRNA-Seq reads to the assembled contigs, *Chlorocebus sabaeus*, dengue virus type 2, and influenza A virus genomes. To determine which contigs were derived from dsRNA, reads that were derived from the forward strand of each contig were extracted using the SAMtools 1.8 [35] samtools view and flag options -f 64 -F16 for Read 1 and -f128 -F16 for Read 2. The number of forward strand reads and the total number of reads that mapped to each contig were then determined using samtools idxstats and used to calculate the percentage of forward strand reads. The strand-specificity of Scriptseq libraries is >98% (Epicentre product literature), therefore contigs with <98% of forward strand reads were considered to be derived from dsRNA rather than possible contaminating ssRNA.

### 2.4. Analysis of dsRNA-Seq and Ribodepleted RNA-Seq Sequences from Mouse Skeletal Muscle Samples

Adaptor and additional sequences flanking library inserts were removed using BBduk.sh of bbmap 38.05 [36], available at https://sourceforge.net/projects/bbmap/, reads were trimmed when average quality score in four base windows fell below 20, and reads smaller than 50 nt were discarded using Trimmomatic 0.36 [30]. Processed dsRNA-Seq reads were mapped to the GRCm38 mouse genome using Bowtie 2.2.9 [34]. The fastq command with the -f 12 option of SAMtools 1.8 [32] was used to extract matched paired reads where both members of the pair did not map to the mouse genome. The reads that did not map to the mouse genome were assembled into contigs at least 750 nt long using Trinity 2.6.6 [31], the longest isoform of related contigs was selected and the contigs derived from dsRNA were determined as described for the Vero samples, except the strand-specificity of Ovation SoLo libraries is >90% (NuGen product literature), therefore contigs with <90% of forward strand reads were considered to be derived from dsRNA rather than possible contaminating ssRNA. To look for similarities between the dsRNA contigs and known sequences, the contigs were used as queries using the BLASTn command of BLAST 2.7.1 [33] against the NCBI nt database. Contigs from each dataset were mapped to Ross River virus strain T48 (GQ433359.1) using the bwa-mem of bwa 0.7.15 [32]. The dsRNA-Seq and ribo-depleted RNA seq reads that aligned to the Ross River virus strain T48 genome were identified using Bowtie 2.2.9 and the no-unal option. Note, the 63 nt poly A tail at the 3′ end of the Ross River virus was deleted for this analysis. Reads that were derived from the positive strand of Ross River virus were extracted using SAMtools 1.3.1 samtools view and flag options -f 64 -F16 for Read 1 and -f128 -F16 for Read 2. Reads derived from the negative strand with flag -f 80 for Read 1 and -f 144 for Read 2. The number of positive strand reads, negative strand reads, and total number of reads were then determined using samtools idxstats and bedgraphs were produced using genomeCoverageBed command of BEDTools 2.25.0 [37]. A similar strategy was used to analyze dsRNA-Seq and ribo-depleted RNA-Seq read mapping to the mouse mitochondrial chromosome.

### 2.5. Analysis of dsRNA-Seq Sequences from Reptile and Deer Tissue Samples

We processed the reptile and deer dsRNA-Seq samples, assembled reads into contigs, and determined which contigs were derived from dsRNA as described for the mouse samples. To investigate similarities between the dsRNA contigs and known sequences, the contigs were used as queries using BLASTn against the entire NCBI nt database, BLASTx againt nr, and BLASTx limited to viral sequences. dsRNA contigs with nucleotide similarity to known viruses were aligned to known virus sequences based on the position of the region of similarity. If contigs only hit to viral proteins, we used sequences as queries in BLASTx and recorded the subject coverage and identity percentage to determine where these contigs were aligning to on the viral genome. We used the same mapping strategy as in the mouse sample analysis in order to align related dsRNA contigs to each other and to display them along viral genomes. RNA-Seq libraries and data analysis for the boa constrictor and chameleon samples were generated as previously described [29]. The *Boa constrictor* genome assembly 6C of the Assemblathon project [38] was downloaded from GigaDB http://gigadb.org/dataset/100060.

### 2.6. Sequence Data Availability

dsRNA and total RNA sequence data are available as raw reads from the NCBI Short Read Archive (SRA) under study accession number SRP201404. Individual accession numbers are Vero_1_dsRNA, SRR9301166; Vero_2_dsRNA, SRR9301167; Vero_3_dsRNA, SRR9301168; Mouse_1_dsRNA, SRR9301169; Mouse_2_dsRNA, SRR9301170; Mouse_3_dsRNA, SRR9301171; Mouse_4_dsRNA, SRR9301172; Mouse_5_dsRNA, SRR9301173; Mouse_1_RNA, SRR9301164; Mouse_2_RNA, SRR9301165; Mouse_3_RNA, SRR9301160; Mouse_4_RNA, SRR9301161; Mouse_5_RNA, SRR9301162; Green_Tree_Python _lung_dsRNA, SRR9301163; Green_Tree_Python_lung_esophagus_dsRNA, SRR9301156; Rough_Scaled_Python_lung_dsRNA, SRR9301157; Boa_Constrictor_kidney_dsRNA, SRR9301158; Veiled_Chameleon_lung_trachea_oral mucosa_dsRNA, SRR9301159; Veiled_Chameleon_lung_liver_kidney_1_dsRNA, SRR9301154; Veiled_Chameleon_lung_liver_kidney_2_dsRNA, SRR9301155; Mule_Deer_brain_dsRNA, SRR9301151; Mule_Deer_lymph_node_dsRNA, SRR9301150; negative_control, SRR9301153; Boa_Constrictor_kidney_RNA, SRR9301152.

## 3. Results

### 3.1. dsRNA-Seq Detects Viral Infections of Cultured Mammalian Cells

As an initial test to determine whether we could detect viral infections in mammalian cells by purifying and sequencing dsRNA, Vero cells were infected with influenza A virus, dengue virus type 2, or were mock infected. Total RNA was isolated from the mock and infected cells and samples were blinded for the remainder of library preparation, sequencing, and data analysis. Immunoblot analysis of the total RNA samples using the anti-dsRNA antibody (Appendix A) revealed that the Vero 2 sample contained a dsRNA species that was not detected in the other samples, which was longer than 5 kb. This observation suggests that the Vero 2 sample was infected with a long single segmented RNA virus. The Vero 3 sample contained a ~2.5 kb dsRNA band not readily detected in the other samples, suggesting that it was also infected with a virus. A ~2 kb band was seen in all three samples, suggesting that it represented a host dsRNA species. These smaller bands were much less abundant than the large dsRNA in the Vero 2 sample (Appendix A), which indicates that the abundance of viral dsRNA (or host dsRNA induced by viral infection) can vary greatly between different viral infections.

dsRNA was purified from the total RNA using a two-step protocol (Figure 1A, see Materials and Methods). First, the total RNA was treated with DNase 1 and a single-strand specific RNase to remove any contaminating DNA and to enrich for double-stranded RNA. The RNase treatment was performed in the presence of 0.2 M monovalent salt to stabilize base pairing interactions to minimize the inadvertent digestion of dsRNA. Subsequently, an antibody that recognizes dsRNA [25,26] was used to immuno-purify the dsRNA. The anti-dsRNA antibody is highly specific for dsRNA, requires at least 40 base pairs of dsRNA for binding, and is sequence-independent, although it does have some preference for binding particular AU-rich sequences [25,26]. Testing this approach using total RNA spiked with varying amounts of in vitro transcribed dsRNA revealed that 1) dsRNA was specifically enriched over single-stranded RNA and 2) the enrichment of the dsRNA was dependent on the anti-dsRNA antibody (Appendix A). Moreover, a broad range of dsRNA amounts can be efficiently isolated. We observed 50–100% recovery of 100 ng to 10 pg of dsRNA (Appendix A).

We isolated dsRNA from the Vero total RNA samples using the two-step purification scheme and estimated by anti-dsRNA immunoblotting that we recovered 50–100% of the dsRNA. We prepared low input cDNA libraries from the dsRNA and sequenced the libraries. All libraries were constructed to maintain strand-specific information, which allowed us to determine if a sequence of interest was present as dsRNA (see Materials and Methods and below).

To mimic a situation where we were trying to identify a virus of unknown sequence, we first assembled the short dsRNA reads into longer contigs (see Materials and Methods) in an attempt to assemble virus genomes. From each of the Vero cell samples we assembled between 39 and 98 contigs that were 500 bases or longer (Figure 1B and Appendix A). Because all three samples, including the mock-infected sample, had similar numbers of contigs, we reasoned that many contigs may be derived from host dsRNA rather than viral dsRNA. Mapping the contigs to the *Chlorocebus sabaeus* (green monkey) genome revealed that the majority of the contigs aligned to the host nuclear or mitochondrial chromosomes (Figure 1B). One possibility is that the host contigs were derived from contaminating single-stranded transcripts. However, when we mapped the reads to the contigs we found that both strands of each contig were represented in the dsRNA reads (see Materials and Methods), consistent with the contigs being derived from host dsRNA (Appendix A). Mapping the dsRNA reads to the *Chlorocebus sabaeus* genome revealed that in the Vero 1 and Vero 3 samples over 90% of the reads were in fact from the host, with the vast majority of the dsRNA coming from the mitochondrial genome (Figure 1C). Thus, dsRNA-Seq reveals the presence of sense/antisense and/or dsRNA in mammalian cells.

To determine if any of the remaining contigs were similar to known viruses, we used BLASTn to look for similarities between the nucleotide sequence of the contigs and sequences in the NCBI nucleotide database. In the Vero 2 sample, six contigs had BLASTn hits to dengue virus type 2 (Figure 1B and Appendix A), a positive sense single-stranded RNA virus with an ~11 kb monopartite genome. These contigs ranged in size from 0.8 to 9.9 kb and represented the entire dengue virus type 2 genome in both orientations except 300 nucleotides at the 5′ end of the virus (Figure 1D). Dengue virus type 2 sequences were very abundant in the dsRNA isolated from the Vero 2 sample, with 76% of the reads mapping to the dengue virus type 2 genome (Figure 1C and Appendix A). The high abundance of viral dsRNA reads relative to host dsRNA in the Vero 2 sample was consistent with the immunoblot analysis, which indicated that this sample had a high abundance of viral dsRNA relative to the other Vero samples (Appendix A).

In the Vero 3 sample, 19 of the contigs shared significant similarities to influenza A virus (Figure 1B and Appendix A). In contrast, the other two samples did not have any contigs with similarities to the influenza A virus (Figure 1B and Appendix A). Influenza A is a negative sense single-stranded RNA virus composed of eight segments. The 19 contigs in the Vero 3 sample ranged in length between 0.5 and 2.3 kb and represented both strands of all eight segments of the virus (Figure 1E). Compared to the high abundance of dengue virus reads in the Vero 2 sample, only 1.47% of the reads in the Vero 3 sample aligned with the influenza A virus (Figure 1C and Appendix A). This result is consistent with the low abundance of viral dsRNA in this sample compared to the Vero 2 sample (Appendix A). Despite only 1.47% of the reads in the Vero 3 sample aligning with the influenza A virus (Figure 1C and Appendix A), we assembled the entire genome, indicating that it is possible to detect viral infection using dsRNA-Seq, even under conditions where the viral RNA is not highly represented relative to the host dsRNA in the recovered dsRNA.

Given that we detected viruses in the Vero 2 and Vero 3 samples, we concluded that the Vero 1 sample was the mock-infected sample. Consistent with this idea, no reads derived from influenza A were detected in the Vero 1 sample. Unexpectedly, short 0.5–0.6 kb contigs derived from dengue virus were assembled in the mock-infected sample (Figure 1B and Appendix A). This was due to the Vero 1 dsRNA sample containing 0.02% reads that mapped to the dengue virus type 2 genome (Figure 1C and Appendix A). These reads are most likely due to contamination from the Vero 2 sample, or by reads mis-assigned due to “index hopping” during sequencing [39].

Uncoding the samples revealed that the infectious agents were accurately identified by dsRNA-Seq, indicating that the enrichment and sequencing of dsRNA is sufficient for the detection of positive and negative sense single-stranded RNA viruses in infected tissue culture cells.

### 3.2. dsRNA-Seq Correctly Detects Viral Infection in Infected Mice

We then asked if we could detect viral infections in animals by isolating and sequencing dsRNA from infected mouse tissue. Viral infection in infected animals or humans could be more challenging given that not every cell in a tissue will be infected, the viral load in infected cells may be low, and the amount of tissue in clinical samples may limit the amount of dsRNA that can be recovered and sequenced.

We obtained five samples of total RNA isolated from infected or uninfected mice. We were blinded to the number of the samples that were infected and uninfected, the type of virus(es) used for infection, and the tissue from which the total RNA was isolated. Using western blot analysis with the anti-dsRNA antibody, we estimated the amount of dsRNA in the samples to be only ~10–120 pg per 1 µg of total RNA (Appendix A). dsRNA was isolated from 5 µg of total RNA (~50–600 pg dsRNA) from each sample and sequenced.

Since the tissue culture experiment suggested that the majority of dsRNA reads would be from the host, we first mapped the dsRNA sequences to the mouse genome. In each dataset, between 78% and 88% of the reads aligned to the mouse genome (Figure 2A, Appendix A). In a second step, the reads that did not map to mouse sequences were assembled into contigs of 750 bases or longer in order to attempt to assemble full-length viral genomes or genome segments. We then used the strand-specific information in our library preparation to determine whether both strands of each putative dsRNA contig were represented in the dsRNA reads. This allowed us to distinguish between contigs derived from dsRNA and those from possible contaminating single-stranded RNA. Four to sixteen contigs derived from dsRNA were assembled in each dataset ranging in length from 0.76 to 23 kb (Appendix A).

To understand the nature of these dsRNA contigs, we asked if they shared similarities to known viral or cellular sequences in the NCBI nucleotide database using BLASTn. We found that four of the five samples had contigs that were greater than 99% identical on the nucleotide level to Ross River virus (Figure 2B and Appendix A). Ross River virus is an alphavirus—a single-stranded positive-sense RNA virus with a single genome segment of ~11.9 kb. The contigs with similarities to Ross River virus represented full-length or nearly full-length viral sequences (Figure 2B). Uncoding the samples revealed that the four RNA samples with dsRNA contigs derived from Ross River virus came from the skeletal muscle tissue of mice at five days post-infection with Ross River virus (T48 strain), with the remaining sample coming from a control mock-infected animal. Thus, dsRNA-Seq correctly identified the virus used for infection and differentiated between infected and non-infected animals.

Given that single-stranded RNA viruses that have not replicated would not be present as dsRNA, dsRNA-Seq should specifically detect ssRNA viruses that are or have been replicated. Consistent with this idea, both positive-sense and negative-sense strands of Ross River virus RNA were well represented (Figure 2C) in the dsRNA-Seq datasets, with 13–43% of the reads that aligned to Ross River virus being derived from the negative strand. Moreover, the high coverage of the negative strand allowed for the assembly of full-length or nearly full-length contigs representing the negative strand of Ross River virus from the infected animals (Figure 2B). Thus, the dsRNA-Seq analysis provided evidence of virus replication in the mouse tissue.

Synthesis of the negative strand of alphaviruses occurs early in infection, followed by a switch to using the negative strand as a template to synthesize full-length genomic RNA, as well as a subgenomic mRNA that encodes the viral structural proteins [40,41]. The mouse samples were collected five days post infection, past the peak of viral replication, therefore dsRNA-Seq is likely detecting negative strands produced earlier in infection. The abundance of reads mapping to the 3′ region of the virus (Figure 2C) is likely due to the high expression of the subgenomic RNA (see below and Discussion), which is synthesized at much higher levels than the full-length genomic RNA [42,43].

Additional contigs were assembled from the dsRNA-Seq reads. The majority of the remaining dsRNA contigs were derived from mouse sequences, primarily from mitochondria, or rRNA sequences from various organisms (Appendix A). Contigs that represent single-stranded sequences (contigs for which reads only mapped to one strand) were also assembled from the reads (Appendix A). Many of these contigs represent sequences from bacteria that are common contaminants in next generation sequencing libraries (Appendix A) [44], which highlights the need to implement strategies to identify such contaminants, particularly when only small amounts of input RNA are available for library preparation [45].

We also identified single-stranded contigs that appeared to be derived from viruses. For example, in all the mouse samples, we assembled contigs that were nearly identical to each other on the nucleotide level (Appendix A) and encode proteins with similarity to picorna-like RNA viruses (Appendix A). These contigs were most likely contaminants of the dsRNA-Seq libraries rather than viruses infecting the mouse samples, given that only their positive strand was represented in the dsRNA-Seq reads (Appendix A). Moreover, reads corresponding to these viruses were not found in ribo-depleted RNA libraries prepared from the same total RNA samples (see below).

### 3.3. Impact of dsRNA-Seq on Detecting RNA Viral Infections

An unanswered question is how dsRNA-Seq compares to sequencing total ribo-depleted RNA for the identification of viruses. To compare the two approaches, we prepared ribo-depleted RNA-Seq libraries from the mouse tissue samples. We first compared the sensitivity of the two methods in detecting the Ross River virus. The percentage of reads that mapped to the Ross River viral genome was decreased in the dsRNA read datasets compared to the ribo-depleted RNA reads (Figure 3A). The percentage of Ross River virus reads was 4–12-fold higher in the traditional RNA-Seq reads than the dsRNA-Seq reads. Thus, conventional sequencing was more sensitive than dsRNA-Seq at detecting the virus.

One reason the Ross River virus sequences were depleted in the dsRNA is that the negative strand of the virus was in low abundance compared to the positive strand in the starting RNA population. Greater than 97% of the ribo-depleted reads that mapped to Ross River virus were derived from the positive strand of the virus (Figure 3B), and the mapping pattern was consistent with most reads deriving from the highly expressed subgenomic RNA which encodes the viral structural proteins (Figure 3B) [42,43]. Reads that covered the negative strand of the virus were present in the ribo-depleted datasets, though at very low abundance relative to the positive strand and within the error rate of the strand-specificity of the library (see Methods). Thus, while the overall abundance of Ross River virus sequences was reduced in the dsRNA-Seq datasets due to the removal of the abundant single-strand positive strand viral RNA during the dsRNA purification, the negative strand of the virus was well represented in the dsRNA-Seq reads (Figure 2C), allowing for the conclusion that the virus replicated in the tissues from which the RNA samples were derived.

A second explanation for the depletion of viral RNA sequences in dsRNA reads relative to the ribo-depleted RNA is that dsRNA-Seq enriches for host mitochondrial dsRNA. In all the dsRNA-Seq datasets but the Mouse 2 sample, 50% or more of the dsRNA reads map to the mitochondrial genome (Figure 3C). In contrast, mitochondrial sequences made up only 7–14% of the ribo-depleted RNA datasets (Figure 3C). The vast majority of the mitochondrial reads in the dsRNA-Seq samples mapped to both strands of a ~1.2 kb region of the mitochondrial (MT) genome, which corresponds to the position of the Nd6 gene (Figure 3D). The Nd6 gene is transcribed from the opposite strand of the mitochondrial genome compared to the other mitochondrial protein-encoding and rRNA genes (Figure 3D). Therefore, the enrichment of mitochondrial sequences in the dsRNA datasets is most likely due to the presence of sense/antisense mitochondrial transcripts from this region.

In summary, although dsRNA-Seq did not lead to enrichment of Ross River viral sequences, it provided evidence for viral replication, which is a potential advantage of dsRNA-Seq. In the case of viruses producing subgenomic RNAs, such as alphaviruses, the presence of these RNAs in conventional RNA-Seq also provides evidence for viral activity. However, dsRNA-Seq is advantageous for the many viruses that only produce genomic length RNA. One way of improving dsRNA-Seq would be to develop ways to deplete the mitochondrial dsRNA sequences, effectively enriching for viral sequences.

### 3.4. dsRNA-Seq Detects RNA Viruses of Multiple Genome Types in Infected Animals

Since we successfully detected RNA viruses in laboratory infected animals, we asked if dsRNA-Seq could do so in naturally infected animals. We obtained nine samples of total RNA isolated from various tissues of infected green tree python (*Morelia viridis*; lung and a mixed lung/esophagus sample), rough scaled python (*Morelia carinata*; lung), boa constrictor (*Boa constrictor*; kidney), veiled chameleon (*Chamaeleo calyptratus*; mixed lung/trachea/oral mucosa and two samples of mixed lung/liver/kidney), and mule deer (*Odocoileus hemionus*; samples from brain and lymph node). These animals were found to be infected via various methods. The green tree and rough scaled pythons died of respiratory disease and were PCR positive for Morelia viridis nidovirus. Standard RNA-Seq confirmed the presence of a snake reptarenavirus and paramyxovirus in the boa constrictor. RNA-Seq was also used to detect a nidovirus infection in the chameleon samples; the chameleon died of respiratory disease. The mule deer was diagnosed with meningoencephalitis on postmortem exam, and was found to be infected with Caprine herpesvirus via DNA-Seq.

We used the same method and platform for dsRNA-Seq to sequence dsRNA isolated from 10 µg of total RNA from each sample. To screen for contaminants, we also processed a negative control sample containing water rather than RNA through the entire dsRNA purification and library preparation procedure. For each sample we obtained ~10–29 million reads (Appendix A).

To mimic a situation where host genomes were unavailable, we skipped host read filtering and directly assembled all reads into dsRNA contigs of 750 bases or longer. We then used the strand specific information to determine which contigs were assembled from dsRNA. A total of 3–87% of contigs were single-stranded possible contaminants, leaving 35 to 1013 contigs derived from dsRNA to analyze per sample, ranging from 0.5 to 23 kb (Appendix A). To determine if the dsRNA contigs were derived from known or related viruses, we used BLASTn and BLASTx to search for similarities between the contigs at the nucleotide and protein level within the entire NCBI nt and nr databases.

Our analysis revealed two general points about the contigs from these samples (Figure 4 and Appendix A). First, a number of dsRNA contigs showed similarities to bacterial sequences, many of which were also found in the negative control and are likely contaminants. Second, in general, the approach depleted the majority of host sequences, with the exception of some mitochondrial dsRNA contigs in the deer samples, which is consistent with overlapping transcription in mammalian mitochondria producing some dsRNA.

More importantly, we were able to identify RNA viruses in the samples from the snakes and chameleon (Table 1).

First, we identified Morelia viridis nidovirus in the green tree and rough scaled python samples. This nidovirus is a single stranded, positive sense ~32.4 kb RNA virus known to cause respiratory disease in pythons [29,46]. Five contigs in the rough scaled python sample and 18 contigs from the green tree python samples shared >90% sequence similarity on the nucleotide level to Morelia viridis nidovirus, and we were able to assemble partial genomes (Figure 5A). Since both strands of the virus were present in the dsRNA-Seq reads, we can conclude these viruses were actively replicating in the animals.

In the boa constrictor tissue, we uncovered a coinfection of at least two distinct reptarenaviruses as well as a reptilian paramyxovirus. Reptarenaviruses contain negative-sense RNA genomes divided into two segments, a small (S ~3.5 kb) and large (L ~7 kb), and co-infections of this type of virus are reportedly common [47,48]. The S segment encodes the glycoprotein precursor (GPC) and the nucleoprotein (NP), whereas the RNA-dependent RNA polymerase (RdRp) and the Z protein (ZP) are encoded by the L segment [47]. Two contigs in this sample were >98% identical at the nucleotide level to the S segment of the University of Helsinki virus (Figure 5B). Fourteen boa constrictor contigs were >90% identical on the nucleotide level and three additional contigs were >96% identical on the protein level to previously sequenced but unclassified reptarenaviruses (Appendix A). These contigs might represent multiple viruses. We identified contigs that mapped to both strands along the entire length of the S and L unclassified reptarenavirus segments (Figure 5C amd Appendix A).

We also found five contigs in the boa constrictor sample that were >90% identical at the nucleotide level and four additional contigs with similarity at the protein level to a number of reptilian paramyxoviruses. (Figure 5D and Appendix A). Paramyxoviruses belong to the family *Paramyxoviridae* and are negative sense, single stranded viruses associated with neuro-respiratory disease in reptiles [49]. We were not able to assemble a complete paramyxovirus genome, although we obtained hits for genes encoding the nucleoprotein, fusion protein, hemagglutinin-neurimidase, and RNA-dependent RNA polymerase (Appendix A). Both the reptarenaviruses and the paramyxovirus contigs had dsRNA-Seq reads that corresponded to both strands of the virus, indicating they had replicated in this boa constrictor.

In addition to confirming viruses in snakes that were previously found via standard RNA-Seq or other methods, dsRNA-Seq allowed us to detect a dsRNA virus in chameleon tissue that was not detected by RNA-Seq. Four contigs in the pooled lung/liver/kidney sample 2 were 76–87% identical on the nucleotide level to a reptilian orthoreovirus, within the family *Reoviridae*, a segmented, dsRNA linear virus that contains ten segments coding for 12–13 proteins. Thirteen additional contigs were 47–95% identical on the protein level to proteins encoded by the same virus (Figure 6A). dsRNA-Seq enriched for this dsRNA virus, given that no reads which align to the orthoreovirus contigs were detected in RNA-Seq data obtained from the same animal. In addition, we detected contigs with similarity on the protein level to reptilian nidoviruses. Eleven contigs in the chameleon pooled lung, trachea, and oral mucosa sample encoded protein sequences that were 26–59% identical to different proteins encoded by snake and lizard nidoviruses (Appendix A). After mapping these contigs to each other and aligning them according to the position of their protein similarity to the reptilian nidoviruses, it appears that they represent the majority of a nidovirus genome (Figure 6B).

We did not identify any viral contigs in the two mule deer samples (brain and lymph node tissue.) Previous DNA-Seq [50] detected a herpesvirus in the mule deer. Herpesvirus is a dsDNA virus, which likely explains our inability to detect viral reads via dsRNA-Seq.

The detection of viral sequences by RNA-Seq from animals for which there are no commercially available rRNA depletion reagents may be limited. Our dsRNA isolation method would be expected to reduce the amount of rRNA sequences and improve the sensitivity of viral detection. We used the boa constrictor sample where we had both dsRNA-Seq and standard RNA-Seq data to examine the sensitivity of the two methods. The boa constrictor genome has not been fully assembled and annotated, however a partial genome assembly is available. When we mapped the boa constrictor reads from both methods to the partial genome assembly we found that 63.7% of the RNA-Seq reads aligned to host sequences (Appendix A). Host sequences were strongly depleted in the dsRNA-Seq, with only 1.1% of the dsRNA reads aligning to the boa genome sequences (Appendix A). To elucidate the efficacy of rRNA depletion by dsRNA-Seq, we compared the percentage of boa constrictor standard RNA-Seq reads and dsRNA-Seq reads that aligned to boa constrictor imperative mitochondrial rRNA sequences (AM236348.1). Mitochondrial rRNA sequences were used in place of nuclear-encoded rRNA because the boa constrictor rRNA sequences have not been annotated. A total of 16.84% of the standard RNA-Seq boa constrictor reads aligned to rRNA, while only 0.47% of dsRNA-Seq reads aligned. The 36-fold reduction in rRNA demonstrates the usefulness of dsRNA-Seq for sequencing organisms without commercially available rRNA depletion reagents.

To determine if the isolation of dsRNA increased the sensitivity of detecting viral sequences, we mapped the dsRNA-Seq and RNA-Seq reads to the assembled unclassified reptarenavirus contigs and found that 10.16% of standard RNA-Seq reads aligned, compared to 42.25% of dsRNA-Seq reads (Appendix A). We performed the same analysis with the University of Helsinki virus contigs and found that 1.32% of standard RNA-Seq reads aligned, compared to 5.28% of dsRNA-Seq reads (Appendix A). For the Anaconda paramyxovirus contigs, 0.05% of standard RNA-Seq aligned, compared to 0.02% of dsRNA-Seq (Appendix A). These results indicate that dsRNA-Seq can be more sensitive than standard RNA-Seq in detecting viruses in naturally infected animals, however, this varies for different viruses, perhaps due to whether they are actively replicating in the tissue examined.

In summary, dsRNA-Seq identified viruses of various RNA genome types in naturally infected animals and allowed us to assemble partial and full-length genomes of viruses infecting snake and chameleon tissue.

## 4. Discussion

By several criteria the two-step method we have developed for purifying dsRNA from total RNA samples is very effective. For example, in the mouse samples, we observed strong depletion of ribosomal RNA and high enrichment of host dsRNA in the dsRNA-Seq libraries. Total RNA from mammalian cells is ~85% rRNA and ~15% mRNA and other RNA species. Based on western blot analysis of total RNA using the anti-dsRNA antibody, we estimated that only ~0.01 to 0.1% of total RNA isolated from mammalian tissue was dsRNA. Given these ratios, if the dsRNA purification method was 99.9% effective at removing rRNA and host single-stranded RNA, we would expect ~42% of the reads in the purified dsRNA to be rRNA and ~50% to be host dsRNA. In all the dsRNA-Seq samples, except for the sample from mouse 2, we observed an even stronger depletion of ribosomal sequences (Figure 3C), indicating that the removal of rRNA was very effective (the nuclease treatment and/or anti-dsRNA immune-purification appears to have been less effective in the mouse 2 sample than the remaining samples). Moreover, there was considerable enrichment of host dsRNA in the dsRNA-Seq libraries, with at least 50% of the dsRNA reads arising from transcripts from opposite strands of the mitochondrial genome (Figure 3C,D). In addition, a global analysis of the alignment of the ribo-depleted and dsRNA-Seq reads to known mouse transcripts revealed that of the dsRNA-Seq reads that aligned to known mouse transcripts, ~50% were derived from the forward strand and ~50% from the reverse strand. In contrast, 90% of the ribo-depleted RNA-Seq reads prepared using the same strand-specific library kit mapped to the forward strand of the known transcripts. This observation is consistent with both strands of the mouse transcripts being present in the purified dsRNA before the strand-specific library was prepared. The isolation of dsRNA by several different strategies has been used to detect viruses in plants [18,51,52], fungi [21], and microbial communities [22]. Our method is likely to be more selective because it employs a two-step purification scheme. In addition, this two-step method should select for RNA that is perfectly base-paired over long regions, such as dsRNA formed from replicative viruses, compared to RNAs that contain imperfectly base-paired secondary structures. Although the anti-dsRNA antibody only requires 40 bp of RNA duplex for binding, the single-strand nuclease treatment would be expected to cleave structures in RNAs that are imperfectly base-paired. During library preparation, which involves heating to 95 °C to dissociate dsRNA, cleavage products from imperfectly base-paired structures would be expected to be too small to be represented in the library.

We posited that dsRNA-Seq should be able to detect all types of RNA viruses, given that dsRNA has been detected in cells infected with positive, negative, and ambisense single-stranded RNA viruses, as well as dsRNA viruses [53,54]. Using dsRNA-Seq, we successfully identified and assembled full or partial genomes for non-segmented and segmented negative- and positive-sense RNA viruses, dsRNA viruses, and as well as ambisense RNA viruses. For the single-stranded RNA viruses, dsRNA-Seq provided evidence that the viruses had replicated in the tissue examined by detecting the positive and negative strand sequences of each virus.

Because viral sequences are generally at low abundance relative to host sequences, one concern when viral sequences are found by high throughput sequencing is whether they represent active viruses or contaminants. The dsRNA-Seq method is useful for distinguishing replicated single-stranded RNA viruses from non-replicating or contaminating viral sequences because it reveals the presence of both viral strands. It should be noted, however, that this method does not necessarily eliminate viruses detected by other methods as potential pathogens, since some viruses may replicate at too low a level to be detected or may not be actively replicating at the time of sampling. One disadvantage of dsRNA-Seq is that it cannot be used to detect single-stranded RNA viruses from clinical samples where viruses would primarily only be present as viral particles, such as serum or cerebral spinal fluid. dsRNA-Seq is particularly useful for sequencing viruses from organisms without sequenced genomes and/or with no commercially available rRNA depletion reagents. Another advantage of this approach over standard sequencing methods is that it may increase the ability to detect dsRNA viruses, such as the dsRNA orthoreovirus we identified in chameleon tissue.

It should be noted that sequences present as “dsRNA” in dsRNA-Seq libraries are not necessarily base-paired with each other in vivo. Hybrids between sense and antisense species during or post RNA isolation would also be expected to be purified as dsRNA. During RNA viral replication and transcription, strands may be separated to limit the detection of dsRNA by the host innate immune system. However, the presence of both viral strands in total RNA should be sufficient to allow the recovery of viral sequences in the dsRNA. Consistent with this idea, there were more dsRNA reads that mapped to the region of Ross River virus which corresponded to the abundant subgenomic RNA (Figure 2C) than to other regions of the virus, presumably because the high abundance of the subgenomic RNA increased the likelihood of there being hybrid molecules of positive and negative strands in this region.

## Figures and Tables

**Figure 1 viruses-11-00943-f001:**
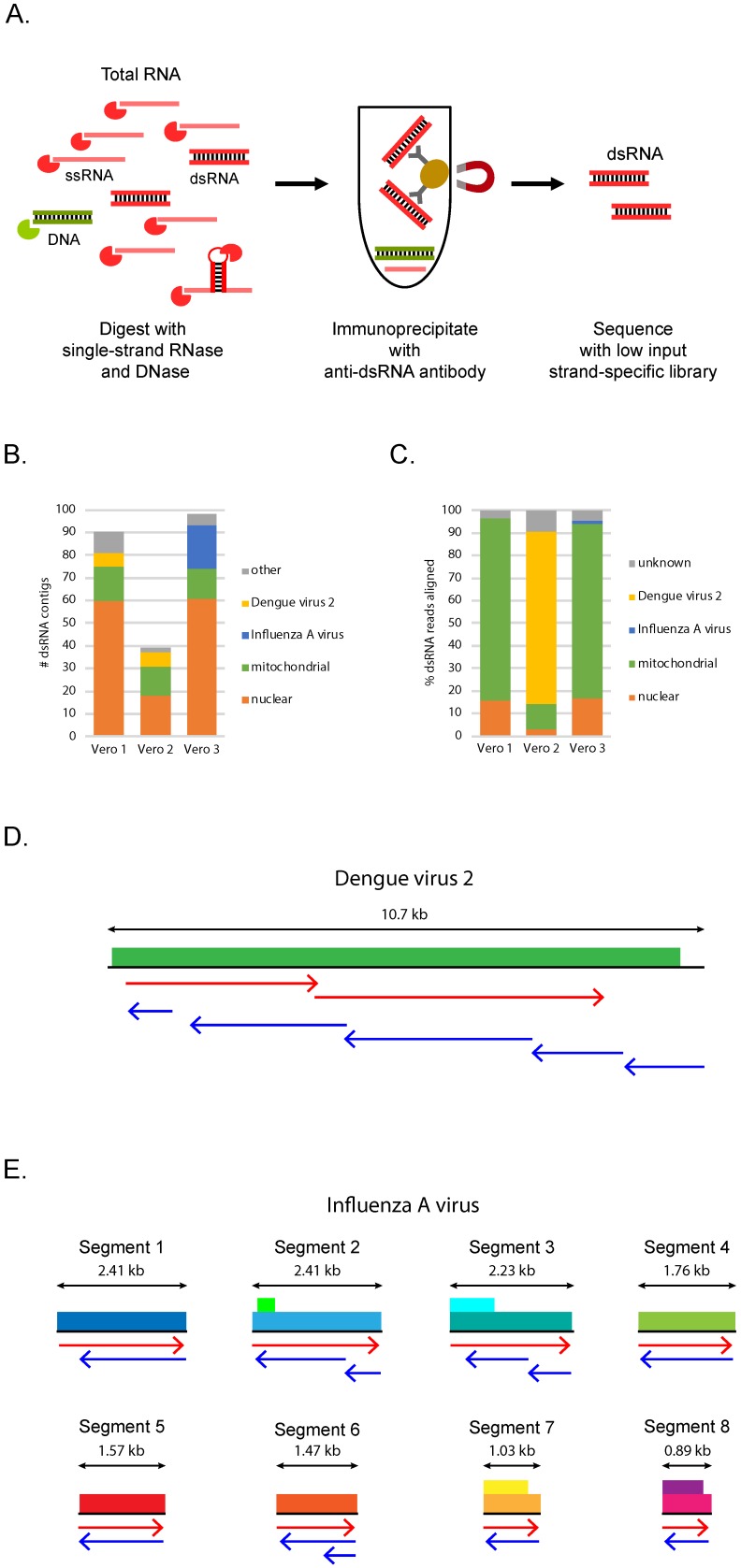
dsRNA-Seq detects viral infections of cultured mammalian cells. (**A**) Outline of the dsRNA purification method; (**B**) number of dsRNA contigs assembled from dsRNA-Seq reads from infected or mock infected Vero cell samples and their classification based on mapping to host nuclear or mitochondrial chromosomes or BLASTn analysis against NCBI nt; (**C**) percentage of dsRNA-Seq reads that align to the host nuclear or mitochondrial chromosomes, influenza A viral genome, dengue virus type 2 genome, or did not align (unknown). For (**D**) and (**E**), viral genomes are illustrated with protein coding regions indicated by colored boxes. Arrows indicate the alignment of contigs to viral genomes or genome segments. Contigs representing the positive strand are in red; negative strand in blue; (**D**) alignment of contigs assembled from Vero 2 sample to dengue virus type 2 genome; (**E**) alignment of contigs assembled from Vero 3 sample to influenza A viral segments.

**Figure 2 viruses-11-00943-f002:**
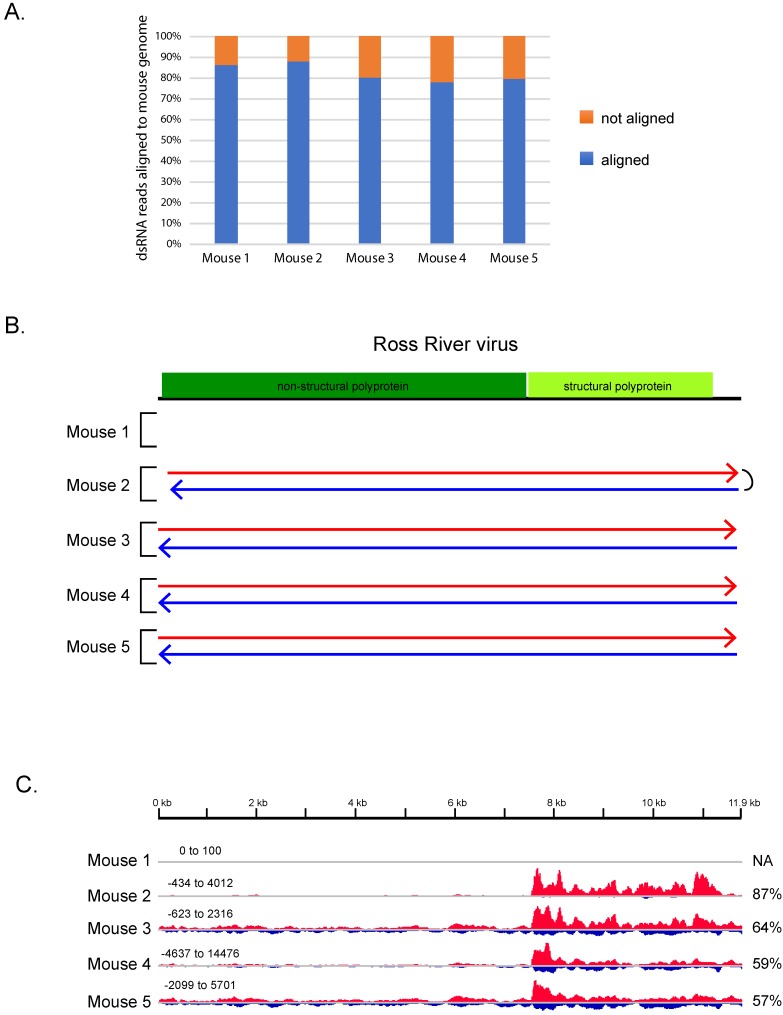
dsRNA-Seq detects viral infection in infected mice. (**A**) Percentage of dsRNA-Seq reads that aligned to the mouse genome; (**B**) Ross River virus with protein coding regions indicated in colored boxes. Arrows indicate the alignment of contigs assembled from dsRNA-Seq reads from mouse samples. Contigs representing the positive strand of the virus are in red; negative strand in blue. The mouse 2 sample contained a single contig that represented both the positive and negative strand of the virus indicated by the link on the right; (**C**) bedgraphs of dsRNA-Seq reads that aligned to Ross River virus genome from mouse samples. Height indicates the base count at each position along the Ross River virus sequence. Red indicates counts on the positive strand. Blue indicates counts on the negative strand. The percentage of the total reads that mapped to the Ross River virus that aligned to the positive strand is indicated on the right. Note: The bedgraphs have not been scaled relative to total number of dsRNA-Seq reads in each sample. Each read set was scaled individually, scale is indicated at top left for each set.

**Figure 3 viruses-11-00943-f003:**
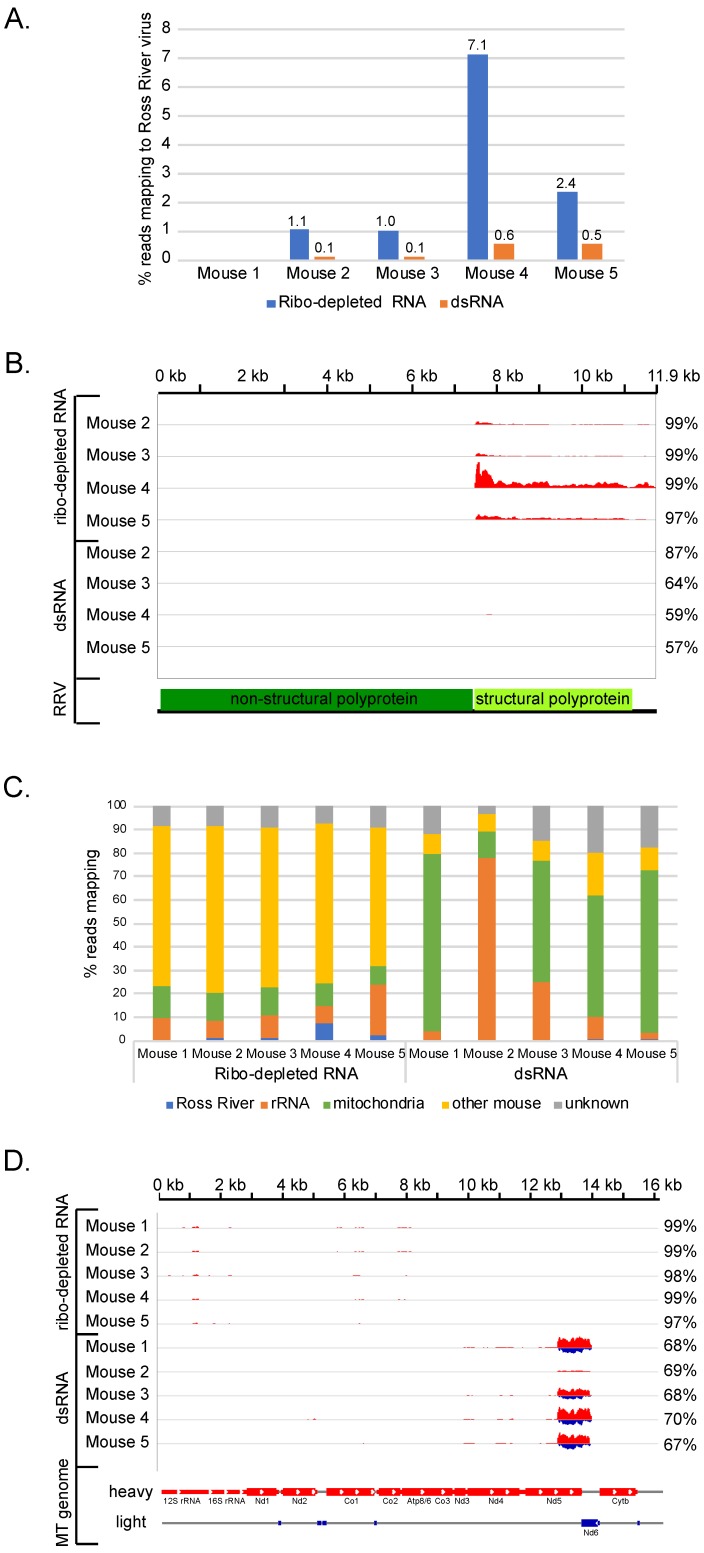
Comparison of dRNA-Seq and ribo-depleted RNA-Seq in identifying virus infections. (**A**) Percentage of reads from ribo-depleted RNA-Seq libraries and dsRNA-Seq libraries prepared from mouse tissue samples that mapped to Ross River virus genome; (**B**) bedgraphs indicating the base count of reads from RNA-Seq libraries and dsRNA-Seq libraries that aligned to each position of the Ross River virus genome. The bedgraphs were scaled relative to the total number of reads in each library and then the entire set was scaled identically as a group. Red indicates counts on positive strand. Blue indicates counts on negative strand. The percentage of the total number of reads that mapped to the Ross River virus that aligned to the positive strand is indicated on the right. A map of the Ross River virus with the location of the regions encoding the non-structural and structural polypeptides is indicated below; (**C**) percentage of reads from ribo-depleted RNA-Seq libraries and dsRNA-Seq libraries from each mouse sample that mapped to the Ross River virus genome, mouse ribosomal sequences, mouse mitochondria genome, other mouse genome sequences, or did not map (unknown); (**D**) bedgraphs indicating the base count of reads from ribo-depleted RNA-Seq libraries and dsRNA-Seq libraries that aligned to each position of the mouse mitochondrial genome. The bedgraphs were scaled relative to the total number of reads in each library and then the entire set was scaled identically as a group. Red indicates counts on positive strand. Blue indicates counts on negative strand. The percentage of total reads in each sample that mapped to the mouse mitochondrial genome that aligned to the positive strand are indicated on the right. A map of the mouse mitochondrial (MT) genome is indicated below, with the genes located on the heavy strand indicated in red and the genes on the light strand indicated in blue.

**Figure 4 viruses-11-00943-f004:**
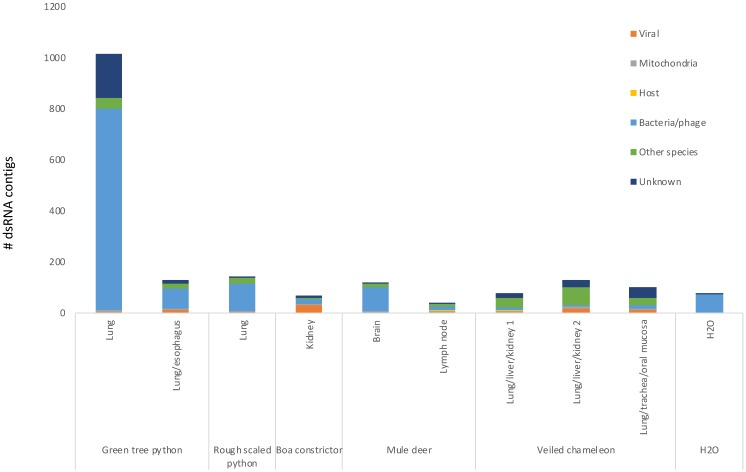
Classification of dsRNA contigs assembled from dsRNA isolated from reptiles and mule deer. Classification of dsRNA contigs from snake, mule deer, and chameleon samples according to BLASTn and BLASTx analysis. Viral contigs that potentially infect the samples; host and mitochondria, contigs with hits to host nuclear or mitochondrial genomes, respectively; bacteria/phage, contigs with hits to bacterial or bacteriophage sequences; other species, contigs with hits to non-host eukaryotic organisms or viruses known to infect non-reptilian species; unknown, contigs without hits to known sequences.

**Figure 5 viruses-11-00943-f005:**
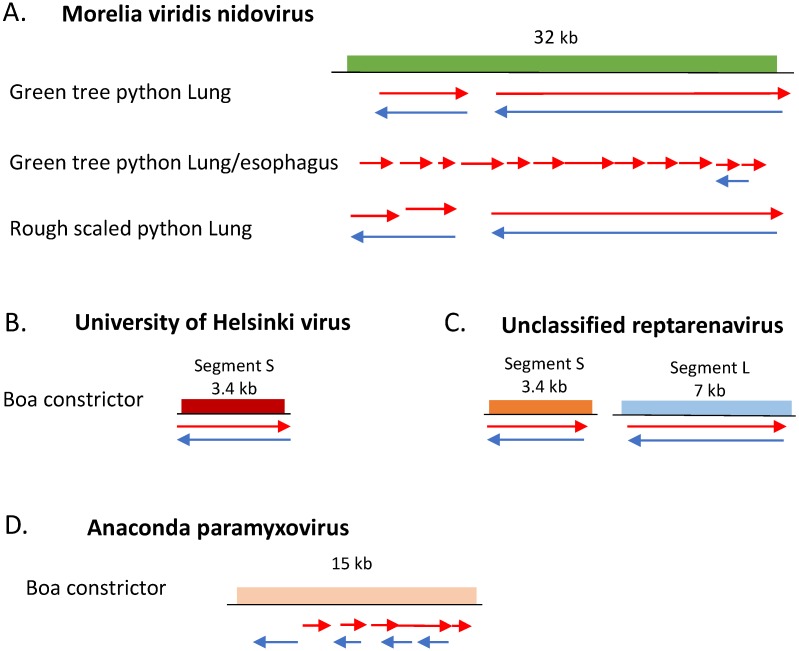
Snake dsRNA contigs aligned to viral genomes. Arrows indicate the relative location of nucleotide sequences in contigs assembled from dsRNA that align with the corresponding virus. Contigs representing the positive strand of the virus are in red. Contigs representing the negative strand in blue. Longest region of contiguous similarity is shown. Colored boxes represent protein coding regions. (**A**) dsRNA contigs isolated from green tree and rough scaled python lung and lung/esophagus pooled tissue mapped to the Morelia viridis nidovirus genome; (**B**) boa constrictor dsRNA contigs mapped to University of Helsinki reptarenavirus S segment; (**C**) representative boa constrictor dsRNA contigs that mapped to entire length of S and L segments of unidentified reptarenavirus strain reptarenavirus/boa constrictor/california/snake38/2009; (**D**) boa constrictor dsRNA contigs mapped to Anaconda paramyxovirus isolate 1110RN047.

**Figure 6 viruses-11-00943-f006:**
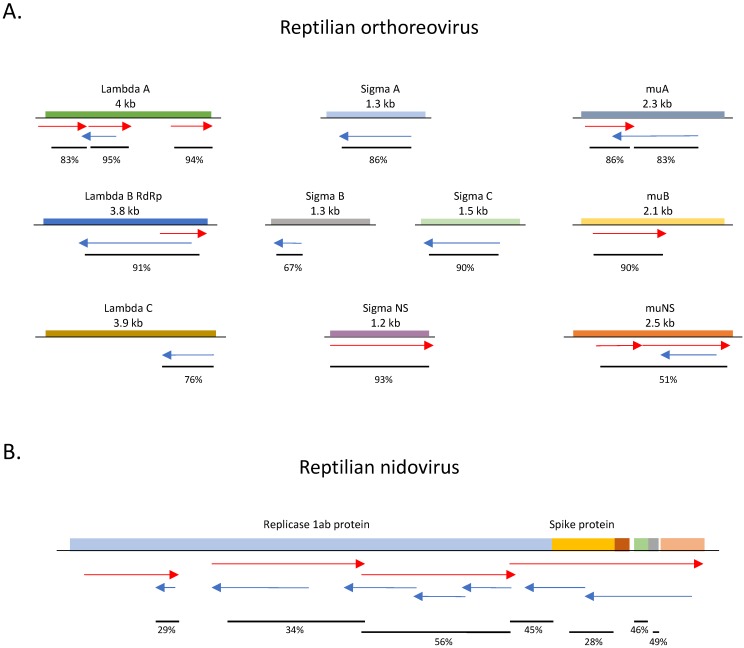
Chameleon dsRNA contigs with protein similarity to known viruses. Diagrams of viruses with coding regions indicated by colored boxes. The locations where contigs shared protein similarities to the virus are indicated by arrows. Contigs representing the positive strand of the virus are displayed by red arrows; the negative strand by blue arrows. The percentage identity between the region in the contig and the corresponding protein coding sequence in the reference virus is indicated below the contigs. (**A**) Schematic of chameleon dsRNA contigs aligned to portions of all ten protein coding segments of reptilian orthoreovirus. The lengths of viral genome segments are indicated. (**B**) Schematic of a reptilian nidovirus displaying the alignment of chameleon dsRNA contigs that had similarity to protein coding sequences of snake and lizard nidoviruses.

**Table 1 viruses-11-00943-t001:** Viruses identified in reptile samples.

Sample	Tissue Type	Virus	Genome Type	Average % Identity	
				Nucleotide ^a^	Protein ^b^
Green Tree Python	lung	Morelia viridis nidovirus	ssRNA (+)	95	93
	lung/esophagus	Morelia viridis nidovirus	ssRNA (+)	94	93
Rough Scaled Python	lung	Morelia viridis nidovirus	ssRNA (+)	94	97
Boa Constrictor	kidney	University of Helsinki virus reptarenavirus	Segmented ssRNA (−)	98	98
		Unidentified reptarenavirus	Segmented ssRNA (−)	97	98
		Reptilian paramixovirus	ssRNA (−)	89	93
Veiled Chameleon	lung/liver/kidney 1	none			
	lung/liver/kidney 2	Reptilian orthoreovirus	Segmented dsRNA	76	85
	lung/trachea/oral mucosa	Reptilian nidoviruses	ssRNA (+)	ND ^c^	43

^a^ average percentage identity of contigs to virus at nucleotide level. ^b^ average percentage identity of predicted protein sequences encoded in contigs to viral protein. ^c^ Not detected.

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
