# Peer review of "dsRNA-Seq: Identification of Viral Infection by Purifying and Sequencing dsRNA"

_viruses, 2019, doi:10.3390/v11100943_

Round 1
Reviewer 1 Report
In this manuscript, Decker et al. describe the performance of a method named “dsRNA-Seq” for RNA virus identification infecting animals. The manuscript describes three main observations. First, the “dsRNA-Seq” can detect viral infections of cultured mammalian cells and experimentally infected mice. Second, “dsRNA-Seq” can also detect RNA viruses in naturally infected reptiles. Third, performances of “dsRNA-Seq” and conventional total ribo-depleted RNA-seq depend on sample, stage of virus infection, etc. Authors also proposed that the important advantage of “dsRNA-Seq” is the ability to identify replicated viral sequences.
The performance of dsRNA sequencing is clear and convincing. However, there are some misleading expressions in this manuscript. Especially, the incorrect expression about the origin of dsRNA and the lack of mention of previously reported dsRNA sequencing methods are critical issues. I think these issues need to be resolved before publication.
The origin of dsRNA is not only a replicative intermediate of ssRNA virus but also the genome of dsRNA virus. Therefore, not all viruses detected by dsRNA specific sequencing are replicated in the target sample. The advantage of “dsRNA-Seq”, proposed by authors, is adaptive for ssRNA virus, but not for dsRNA virus.
NGS methods to determine the sequence of dsRNAs have been reported by Roossinck et al., Yanagisawa et al. and Urayama et al. These reports are essential to understand the importance of this study. Hence, the lack of a mention is not fair. Urayama et al. also reported complete genome sequence of a novel RNA virus from deep-sea animal.
Roossinck, Marilyn J., et al. "Ecogenomics: using massively parallel pyrosequencing to understand virus ecology." Molecular Ecology 19 (2010): 81-88.
Yanagisawa, Hironobu, et al. "Combined DECS analysis and next-generation sequencing enable efficient detection of novel plant RNA viruses." Viruses 8.3 (2016): 70.
Urayama, Syun‐ichi, et al. "Unveiling the RNA virosphere associated with marine microorganisms." Molecular ecology resources 18.6 (2018): 1444-1455.
Urayama, Syun-ichi, et al. "Complete Genome Sequence of a Novel RNA Virus Identified from a Deep-Sea Animal, Osedax japonicus." Microbes and environments (2018): ME18089.
Major
Line 21-22, 31> As described above, viruses detected by “dsRNA-Seq” include the genome of dsRNA virus. The author cannot conclude that the virus obtained by “dsRNA-Seq” is replicating in the target sample. For the same reason, notification of [noninfectious viral particles or contaminants] is available only for ssRNA virus.
Line 29> In this study, no virus which has [no similarity to known viruses] is identified.
Line 621-631> It is overestimation for the potential ability of “dsRNA-Seq”. About the identification of virus which has [no similarity to known viruses], the only advance of “dsRNA-Seq” is that the sequence, obtained by this method, likely to originate from dsRNA, compared with the conventional total ribo-depleted RNA-seq. However, most of dsRNA contigs were originated from non-virus source (Figure 1B). Hence, “dsRNA-Seq” does not provide sufficient information for identification of virus which has “no similarity to known viruses”, even when genomic information of host species is available. To discuss this potential ability, the more methodological progress is required.
Some of the important information are not shown for measuring the performance of “dsRNA-Seq”.
Line 556-564> Those values are apparently high compared with the case of Figure 3C. Please show the percentage (estimated without reference genome) of origins of reads like as Figure 3C.
Figure 3A> The value for Mouse2 and 3 might be too small to represent as a figure. Please add the values.
Line 574> Theoretically, it is difficult to determine full-length genome by “dsRNA-Seq” without a reference genome of related virus. To avoid misunderstandings, it should be clearly stated in the paper that a reference sequence is required to determine the full-length sequence.
Minor
Figure 6A> Please mention about the biased directions of obtained contigs. Why are most of the reptilian orthoreovirus contigs derived from one direction? In Figure 1,2 and 5, bi-directional contigs were detected.
Line 212> Should be “broad range of dsRNA concentration”
Line 361> There is no Figure S3C.
To the best of my knowledge, detection of negative-stranded RNA virus is great progress, because it is thought that less mount of dsRNA is formed by negative-strand RNA viruses in the host cell. Therefore, to discuss the origin of negative-stranded RNA virus-derived dsRNA should be valuable.
Weber, Friedemann, et al. "Double-stranded RNA is produced by positive-strand RNA viruses and DNA viruses but not in detectable amounts by negative-strand RNA viruses." Journal of virology 80.10 (2006): 5059-5064.
Reviewer 2 Report
The ms describes a new method based on dsRNA sequencing for the high throughput identification of RNA viruses in various samples. The technique has its merit above standard RNA seq approaches as it is particularly sensitive for the detection of the dsRNA form of replicating viruses. And the authors tested their tools on a variety of samples. However, it is clear that this aproach has its own limitation as well that should be discussed a bit further. One major concern is the coverage of only 500-600 nts for some cases (and the authors missed to explain what could be the limiting factors in such short coverage). With such small coverage it is unclear how the authors could claim that with dsRNA sequencing they could assemble new unknown viruses. While they provided evidence of identification of full length viruses in some instance, they should modify their abstract (line 23) to include also partial genome and the potential limiting factors linked to the nature of the virus to be tested.
Main comments:
It is unclear how do the authors distinguish dsRNA generated from RNA secondary structure from the virus replicative form.
Line 199: how many days after infection the total RNA was extracted. how does the level of infection determine efficacy of the recovery of dsRNA?
As they mentioned in line 254, only 1.47% of the reads aligned to the virus. What are the reasons of such low efficiency? Was it related to viral replication? Did the authors used other techniques for verifying efficiency of infection.
Line 256 I think the authors should rephrase the sentence “under conditions were the viral RNA is not highly represented” because this low recovery can also be due to limitation of the dsRNA technique and the authors should have compared the approach with RNAseq alone.
Line 269: the short coverage is one major concern since dengue virus RNA should be about 10000 bases. It is unclear what are the factors that led to such limited coverage. Is it related to low infectivity of the sample? Have the authors verified with other detection tools? Or is it a limitation of the dsRNA sequencing?
Line 272 and 361: it is unclear how do the authors differentiate “contamination’ vs actual read out but with low efficiency.
Line 364: the conclusion needs to be modified as the authors did not do RNAseq to compare the initial viral RNA sequence with the replicating level.
Line 564: what are the factors that control variation depending on the virus nature?
Reviewer 3 Report
Decker et al. report the application of HTS on dsRNA template for the identification of viral infections. The paper is well-written, applied procedures are appropriate and results are coherent with the adopted methodology. In view of that, I recommend its’ publication in Viruses.
However, it seems to me that authors have somehow self-attributed the application of the Next Generation Sequencing technology on dsRNA template as their own novelty or distinction, and didn’t mention that this policy of using such templates to identify known and unknow viral sequences has been adopted in the past (almost exclusively by plant virologists). Hereafter, are some examples found in the literature that they should be reported in their introductory section.
Marais A, Faure C, Bergey B, Candresse T. (2018). Viral Double-Stranded RNAs (dsRNAs) from Plants: Alternative Nucleic Acid Substrates for High-Throughput Sequencing. Methods Mol Biol. 2018;1746:45-53. doi: 10.1007/978-1-4939-7683-6_4.
Elbeaino T., Digiaro M., Uppala M., Sudini H. (2015). Deep-sequencing of dsRNAs recovered from mosaic-diseased pigeonpea (Cajanus cajan L.) revealed the presence of a novel emaravirus: Pigeonpea sterility mosaic virus 2. Archives of Virology, 160 (8): 2019-2029. doi: 10.1007/s00705-015-2479-y.
Round 2
Reviewer 1 Report
I have no more major concerns.
Minor
Title> In this manuscript, the authors used only animal samples. Therefore, the title should include “animal” or related word.
Please check the use of “double stranded RNA” and “dsRNA”. For example, Line 74 is not the first use of “dsRNA”.
Line 355> “5’” Is this collect?
Author Response
Title> In this manuscript, the authors used only animal samples. Therefore, the title should include “animal” or related word.
While we appreciate that this manuscript only describes work with animal infections, the approach is generally applicable to viral identification in other organisms. Indeed, we have used the same approach on microbiobial communities with great success (manuscript in preparation). Given these points, we have chosen not to change the title.
Please check the use of “double stranded RNA” and “dsRNA”. For example, Line 74 is not the first use of “dsRNA”.
Thank you for pointing this out. We have revised the manuscript to define dsRNA and ssRNA at first mention in the Abstract.
Line 355> “5’” Is this collect?
It is correct that the assembled contigs did not include sequences at the 5’ end of the dengue virus type 2 genome. We have revised the sentence to on line 355 to communicate this observation more clearly.
Original sentence
These contigs ranged in size from 0.8-9.9 kb and represented the entire dengue virus type 2 genome in both orientations except the 5’ most 300 nucleotides (Figure 1D).
Revised
These contigs ranged in size from 0.8-9.9 kb and represented the entire dengue virus type 2 genome in both orientations except 300 nucleotides at the 5’ end of the virus (Figure 1D).